# Renal Complications Related to Checkpoint Inhibitors: Diagnostic and Therapeutic Strategies

**DOI:** 10.3390/diagnostics11071187

**Published:** 2021-06-30

**Authors:** Julie Belliere, Julien Mazieres, Nicolas Meyer, Leila Chebane, Fabien Despas

**Affiliations:** 1Department of Nephrology and Organ Transplantation, University Hospital of Toulouse, 31 400 Toulouse, France; 2INSERM U1048 (Institute of Metabolic and Cardiovascular Diseases), 31 400 Toulouse, France; 3Department of Biological Sciences, Paul Sabatier University, 31 400 Toulouse, France; mazieres.j@chu-toulouse.fr (J.M.); meyer.n@chu-toulouse.fr (N.M.); 4Institut Universitaire du Cancer Toulouse Oncopole, 31 400 Toulouse, France; 5Department of Pneumology, University Hospital of Toulouse, 31 400 Toulouse, France; 6Department of Dermatology, University Hospital of Toulouse, 31 400 Toulouse, France; 7Service Pharmacologie Médicale et Clinique, Centre Midi-Pyrénées de PharmacoVigilance, de Pharmacoépidémiologie et d’Informations sur le Médicament, 31 400 Toulouse, France; Leila.chebane@univ-tlse3.fr (L.C.); fabien.despas@univ-tlse3.fr (F.D.); 8Service de Pharmacologie Médicale et Clinique, Faculté de Médecine, Université Paul Sabatier, Equipe PEPSS Centre d’Investigation Clinique 1436, INSERM 1297, 31 400 Toulouse, France

**Keywords:** renal complications, acute kidney injury, immune check point inhibitors, immune-related adverse events

## Abstract

Immune checkpoint inhibitors (ICI) targeting CTLA-4 and the PD-1/PD-L1 axis have unprecedentedly improved global prognosis in several types of cancers. However, they are associated with the occurrence of immune-related adverse events. Despite their low incidence, renal complications can interfere with the oncologic strategy. The breaking of peripheral tolerance and the emergence of auto- or drug-reactive T-cells are the main pathophysiological hypotheses to explain renal complications after ICI exposure. ICIs can induce a large spectrum of renal symptoms with variable severity (from isolated electrolyte disorders to dialysis-dependent acute kidney injury (AKI)) and presentation (acute tubule-interstitial nephritis in >90% of cases and a minority of glomerular diseases). In this review, the current trends in diagnosis and treatment strategies are summarized. The diagnosis of ICI-related renal complications requires special steps to avoid confounding factors, identify known risk factors (lower baseline estimated glomerular filtration rate, proton pump inhibitor use, and combination ICI therapy), and prove ICI causality, even after long-term exposure (weeks to months). A kidney biopsy should be performed as soon as possible. The treatment strategies rely on ICI discontinuation as well as co-medications, corticosteroids for 2 months, and tailored immunosuppressive drugs when renal response is not achieved.

## 1. Introduction

Immune checkpoint inhibitors (ICIs) have been approved in the field of oncology, providing an original antitumor approach compared to chemotherapies. Their utilization relies on the drug’s capacity to repair dysfunctional T cells resulting in the regression of various cancers. The “price to pay” is the risk of autoimmunity, leading to immune-related adverse events (irAEs) and, in some cases, end organ damage. The contributions of ICIs to kidney toxicity have been neglected and underestimated for several years, but it has now been acknowledged that they lead to acute kidney injury (AKI). This impacts renal function and, subsequently, oncologic treatment choices must be weighed. This review focuses on diagnostic and therapeutic strategies for ICI-related renal complications.

### 1.1. ICIs

Both cytotoxic T-lymphocyte-associated protein 4 (CTLA-4) and programmed death 1 (PD-1) play a role as physiologic brakes on unrestrained cytotoxic T-effector function. CTLA-4 (CD 152) is a member of the B7/CD28 family. It mediates immunosuppression by indirectly diminishing signaling through the co-stimulatory receptor CD28. The CTLA4 blockade also restores T cell three-signal activation. Ipilimumab is the first and only FDA-approved CTLA-4 inhibitor. PD-1 is an inhibitory transmembrane protein expressed in T cells, B cells, natural killer cells, and myeloid-derived suppressor cells. Programmed death-ligand 1 (PD-L1) is expressed on the surface of multiple tissue types, including many tumor cells and hematopoietic cells. PD-L2 is more restricted to hematopoietic cells. A blockade of the PD-1/PDL-1 pathway can enhance antitumor T cell reactivity and promote immune control over cancerous cells. Since the FDA approval of ipilimumab (human IgG1 k anti-CTLA-4 monoclonal antibody) in 2011, eight more ICIs have been approved for cancer therapy. PD-1 inhibitors (pembrolizumab, nivolumab, cemiplimab) and PD-L1 inhibitors (atezolizumab, avelumab, and durvalumab) are on the current list of approved agents [1]. Recent anti-CTLA4 antibodies such as tremelimumab and quavonlimab (MK-1308) are now used in combination with anti-PDL1. For example, a combination of the anti-CTLA4 tremelimumab and the anti-PDL1 durvalumab is promising in advanced non-small cell lung cancer [2], head and neck squamous cell carcinoma [3], and other solid tumors such as advanced hepatocellular carcinoma [4]. The use of quavonlimab in combination with pembrolizumab in first-line treatment has also been reported in advanced non-small-cell lung cancer [5] and advanced small-cell lung cancer [6]. Recent studies have identified several new immune checkpoint targets, such as lymphocyte activation gene-3 (LAG-3), T cell immunoglobulin and mucin-domain containing-3 (TIM-3), T cell immunoglobulin and ITIM domain (TIGIT), and V-domain Ig suppressor of T cell activation (VISTA) [7]. The studies have generated promising results in clinical trials. As reported in Table 1, the number of ICIs is increasing.

While therapy with this class of agents has resulted in improved clinical outcomes for patients with multiple tumor types, a broad spectrum of irAEs may affect any organ system, with variable clinical presentations.

### 1.2. Incidence of Renal irAEs

Although severe irAEs remain rare (~10% of the cases under monotherapy), they can become life-threatening if not anticipated and managed appropriately [8]. The highest frequency has been observed with CTLA4 antibodies and combinations of ICIs. Global grade III and IV toxicities occur in 20% of patients. Renal toxicities are not the most frequent [9]: the incidence of AKI is 2% for ipilimumab, 1.9% for nivolumab, 1.4% for pembrolizumab, and 4.9% for the ipilimumab and nivolumab combination [10], but it is hypothesized that it will rise to between 9.9 and 29% in the near future [11]. The proportion of renal irAEs has not yet been detailed, which is why we carried out a search on the VigiBase Pharmacovigilance database. In February 2021, VigiBase^®^ contained >24 million individual case safety reports (ICSRs) from 127 countries. Each ICSR consists of a description of the drugs that are suspected to cause adverse drug reactions and contains information on patient age, gender, medical history, country, drugs taken, and drug initiation and stop dates. As reported in Table 2, the proportion of renal ICSRs (classified by the System Organ Class “Renal and Urinary Disorders”) ranged from 2.6 to 7.9%.

The Uppsala Monitoring Centre (UMC) receives individual case safety reports (ICSRs) of suspected ADRs sent by national pharmacovigilance centers, which are stored in the World Health Organization’s (WHO) global safety database (VigiBase^®^). In February 2021, VigiBase^®^ contained >24 million ICSRs from 127 countries. Each ICSR consists of a description of the drugs that are suspected of causing ADRs and contains information on patient age, gender, medical history, country, drugs taken, and drug initiation and stop dates. Drugs are coded using the WHO drug dictionary, covering over 150,000 medicines and vaccines. The distribution of ICSRs is based on pharmacovigilance notifications sent by practitioners or patients. Therefore, the frequencies in the table are different from the overall incidence of adverse drugs reactions evaluated during clinical trials. The percentages in the table are used to assess the adverse drug reaction profile of each drug.

The distribution of ICSRs is based on pharmacovigilance notifications sent by practitioners or patients. Therefore, the frequencies in the table are different from the overall incidence of adverse drugs reactions evaluated during clinical trials. The percentages in the table are used to assess the adverse drug reaction profile of each drug. *Eftilagim, no ICSR reported in VigiBase as of 11 June 2021.

### 1.3. A Paradigm Shift from Renal “Toxicity”

Contrary to conventional chemotherapies, ICIs can lead to renal injury through various mechanisms. When describing renal complications related to ICI exposure, one should be aware of a novel paradigm that involves deleterious indirect immune responses as opposed to direct toxicity, which is the case for numerous anticancer molecules [12].

There is no evidence of a dose–response relationship. Contrary to other drugs, ICIs are not excreted by glomerular filtration. They display the same pharmacokinetic properties as other therapeutic antibodies, which include little impact of kidney or liver function impairment. The dominant mechanism of ICI clearance remains proteolytic catabolism [13]. ICIs are distributed by means of diffusion and convection within tissues. The neonatal Fc receptor is responsible for the transport of ICIs back into the vascular system, which prevents the intracellular degradation of these drugs and, consequently, prolongs their half-life [14]. On the other hand, the generation of antibodies against ICIs increases clearance as well as receptor-mediated endocytosis. That is why the half-lives of ICIs are also quite long (6–27 days) and are affected by immune system determinants that increase interindividual variability [13].

Renal complications are mediated by immune responses with individual determinants. In fact, ICIs impact peripheral tolerance. Whereas CTLA4 signaling occurs in the tumor-draining lymph nodes, PD1/PDL1 blockade occurs at the tissue level and in the tumor microenvironment. A recent special review [15] extensively describes several fundamental hypotheses evoked to explain ICI-related renal toxicities, including the implication of gut microbiome and immunosenescence pathways. As shown in Figure 1, (i) checkpoint inhibition could lead to the production of autoantibodies against self-antigens that share epitopes with tumors. This has been described for a lupus-like nephropathy that occurs after ipilimumab administration [16]. (ii) Checkpoint inhibition could drive the activity of self-reactive T cell clones. This was previously described in a case report of a patient presenting with fulminant myocarditis in which the selective clonal T cell populations infiltrating the myocardium were identical to those in tumors and skeletal muscle [17]. Regarding the kidneys, renal tubular cells express PDL1, which protects them from T cell-mediated autoimmunity. In fact, PDL1 is constitutively expressed on human cell line HK-2 cells and is dramatically up-regulated by inflammatory signaling by IFN-gamma for example [18]. Furthermore, PD-L1 is frequently expressed in various renal pathologies unrelated to ICI therapy and could be a prerequisite for susceptibility to developing AKI and deleterious immune-related AIN [19]. In addition, ICI-related nephritis is a rare event in renal cell carcinoma, but it may portend a higher likelihood of response. One possible explanation is antigenic overlap between normal tubular cells and tumor cells [20]. (iii) ICIs could lead to the reactivation of drug-specific T cells. The loss of tolerance for common drugs such as proton-pump inhibitors is suspected [21]. In brief, ICIs disrupt the peripheral immune tolerance between tubular cells, dormant auto-reactive T cells, and tolerogenic dendritic cells and promote the migration and activation of effector T cells in renal tissue. ICIs are also known to participate in pro-inflammatory cytokine release (mainly CXCL-10, TNF-alpha, and IL-6). This is why renal toxicity is very hard to predict and can occur after a long period.

The renal complications of ICIs encompass a wide landscape. In a multicentric study of 138 patients with ICI-associated AKI, defined as a two-fold increase in serum creatinine or dialysis requirement directly attributed to ICIs, acute tubulointerstitial nephritis (ATIN) was the dominant lesion in 93% of the 60 patients biopsied [22]. However, glomerular lesions have also been reported recently [23]: 45 cases of biopsy-confirmed ICI-associated glomerular disease were identified. Several lesion types were observed, the most frequent being pauci-immune glomerulonephritis (GN) and renal vasculitis (27%) [24], podocytopathies (24%) (minimal change disease [25], or FSGS [26]), and complement 3 GN (C3GN; 11%). Concomitant AIN was reported in 41% of patients. Other glomerular lesions have been observed [27], including IgA-associated glomerulonephritis (GN) [28], Goodpasture syndrome [29], membranoproliferative GN [30], lupus-like nephropathy [16], and thrombotic micro-angiopathy [31]. Furthermore, an overlap between ATIN and glomerular diseases can be noted (Figure 1). Interestingly, in some patients, renal lesions were only revealed by electrolyte disorders, including hyponatremia secondary to hypophysitis, hypokalemia [32], and distal renal tubular acidosis [33].

## 2. Diagnostic Strategies

Renal complications do not necessarily mean AKI. In some patients, ICI-related toxicity is revealed by only mild abnormalities such as isolated electrolyte disorders, or isolated urinalysis abnormalities (e.g., a single low-grade proteinuria at the early phase of podocyte injury). As patients with cancers have decreased muscle mass, and unlike conventional chemotherapeutics, ICIs do not cause classical drug nephrotoxicity, diagnosing ICI-related renal complications may be difficult. First of all, the conventional oncological approach with grade III to IV irAEs is inapplicable to renal function assessment. In the 2018 oncology guidelines [34], the interruption of ICIs and a consultation with the nephrologist were recommended when serum creatinine increased by a factor of 2–3. It has been noted that even a rise that is <1.5 could be meaningful. However, according to the KDIGO staging and definition system for acute kidney injury, a 0.3 mg/dL increase within 48 hours is enough to assess AKI stage one [12] and to refer the patient to a nephrologist. The earlier the diagnosis of nephritis is made, the greater the chance of success with corticosteroids is.

We will now focus on the most frequent clinical situation, that is to say, the diagnosis of AKI in a patient with a history of ICI exposure, and illustrate the diagnostic strategy in a three-step process, summarized in Figure 2.

### 2.1. First Step: Assessment for Clinical Renal Presentation

The diagnostic strategy includes precise screening for the medical history (nephrological and oncological aspects), current and previous medications, as well as cardiovascular risk factors and habits. The patient has to be precisely questioned on recent events such as contrast CT scan, nephrotoxic drug exposure (angiotensin-converting enzyme inhibitor, angiotensin receptor blocker, non-steroidal anti-inflammatory drugs, proton pump inhibitors, bisphosphonates), dehydration, and screening for a systemic disorder (Sjögren’s syndrome, Raynaud’s syndrome, arthralgia, fever, digestive disorders, chest pain, urinary disorders).

Clinical examination focuses on signs of dehydration and possible extrarenal irAEs, especially cutaneous lesions. An extrarenal irAE, most often a rash, developed before or concomitant with AKI in 43% of the cases in a recent series [22].

Biological exams include a urinalysis for leukocyturia, hematuria, urine culture, as well as sodium/potassium ratio, magnesium, and sodium excretion fraction calculation, proteinuria, micro-albuminuria, and urine creatinine measurement. Given that some cases of ICI-related kidney toxicity may be restricted to isolated electrolyte disorders, clinicians should be aware of small variations in routine lab tests that suggest tubular dysfunction. A cystatin C measurement could be very useful to confirm a decrease in the estimated glomerular filtration rate (eGFR) in patients receiving treatments that cause inhibition of renal transporters leading to a reversible and dose-dependent increase in creatinine [35]. In a series by Cortazar et al. on 138 patients, the urine protein-to-creatinine ratio was >0.3 g/g in 71% of the patients, urine dipstick was positive for leukocyte esterase, and pyuria was noted on the urine sediment in approximately half of the patients. None of these characteristics differed significantly according to AKI severity [22].

There should then be a systematic screening for the following other irAEs: thyroid disorders (TSH dosage), electrocardiogram, troponin, and BNP levels for cardiac injury, liver enzyme test, RBCs for associated hematological abnormalities (thrombocytopenia, TMA, hypereosinophilia), CPK (rhabdomyolysis), electrophoresis and immunofixation, and cytokine profiling (if available in the context of COVID). An ICI dosage should be performed if possible [36].

Finally, kidney imaging should be performed (echography or CT scan without contrast).

At this point, the clinician should be able to determine whether the patient suffers from pre-renal, post-renal, or intrinsic AKI and which renal compartment is involved in the ICI toxicity (tubulointerstitial, glomerular, or vascular origin).

### 2.2. Second Step: In-Depth Assessment for Suspected Lesions

If an ICI-related renal complication is suspected, the benefit/risk ratio for kidney biopsy for histological analysis should be discussed, with a multidisciplinary approach if possible, involving both oncologists and nephrologists. This discussion takes into account the feasibility of the procedure (kidney size, accessibility, coagulation disorders, anti-coagulation retrieval), the patient’s general health status, their choice and feelings about the overall situation, and most importantly, the possible therapeutic changes if the kidney biopsy is performed.

Whereas international guidelines do not recommend discussing kidney biopsy as a first-line investigative tool [34,37], nephrologists oppose this practice. In a recent case series, five out of ten patients with suspected ICI-related AKI were found to have acute tubular injury/necrosis on biopsy [38]. This underlines the importance of proving acute tubular necrosis without an inflammatory component to avoid exposure to steroids. Furthermore, the identification of an associated lesion (e.g., a glomerular lesion such as vasculitis) as well as typing of immune infiltrates can impact the treatment choice to preserve future kidney function. Finally, a kidney biopsy provides precious information on baseline renal parenchyma through which a prognosis can be made. A kidney biopsy should be as representative as possible (including fixed and frozen sections, with a sufficient number of glomeruli) to allow routine staining and the detection of antibodies. Immunophenotyping of the immune infiltration in the kidney is mandatory to exclude lymphoproliferative disorders. Screening for T cell clones in the kidney could also be useful in some cases and requires a paraformaldehyde fixation to optimize their detection. T cell clones can be suspected in case of a positive membrane surface marker analysis (CD4, CD8). The pathologist should be able to specify whether the infiltrate is monomorphic or polymorphic. However, the confirmation of T cell clones depends on the results of a T-cell receptor gene rearrangement study. Expensive approaches such as ImmunoSEQ technologies can be considered. Kappa/lambda restriction orients more toward B cell clones.

When a kidney biopsy is not possible, non-invasive markers have been studied in preliminary works. Currently, there is no formal recommendation concerning this. Soluble urinary CD163 (suCD163) appears to be promising to reflect intra-renal infiltration by macrophages [15]. In fact, in a series of 72 cases of biopsy-proven acute tubular necrosis (ATN), an older age and a higher density of CD163+ macrophages predicted non-recovery, whereas the AKI stage, tubular injury score, and the density of CD68+ macrophage cells did not. The density of CD163+ M2 macrophages was an independent predictor of low eGFR at 3 months in advanced-stage AKI [39]. In ICI-related ATIN, some authors report the presence of CD163+ macrophages in kidney immune cell infiltration [40,41]. In various diseases associated with AKI, such as vasculitis and lupus nephropathies [42,43], suCD163 has been identified as a relevant marker. Prospective studies are needed to assess whether it plays a similar role in ICI-related AKI, especially when an invasive procedure is not possible. Recently, an increase in 18F-flourodeoxyglucose uptake in the renal cortex in a patient with checkpoint inhibitor-associated acute interstitial nephritis was described in a case report. This suggests that 18F-flourodeoxyglucose positron-emission tomography could be a valuable diagnostic test for immune-mediated nephritis, particularly in patients where a timely kidney biopsy is not feasible [44].

### 2.3. Third Step: Assessment for ICI Causality in Renal Lesions

For frequent irAEs (hepatitis, colitis, and pneumonitis), a dedicated inpatient immune toxicity service (ITox) was established for patients admitted with irAEs using internal guidelines based on those of the NCCN and ASCO. Algorithms for defining an irAE as “definite”, “likely”, “possible”, or “unlikely” were developed [45]. Unfortunately, such an approach has not yet been deployed for renal irAEs and it is the practitioner’s responsibility to make this determination. Assessing the causality between the adverse events and suspected drugs is the most challenging task in pharmacovigilance. It requires close consideration of both the irAE and the suspected ICI, as well as patient-related factors, suspected concurrent drugs, and other medical conditions of the patient. Though different methods were developed to assess causality, no single method has been proven to produce an accurate or authentic ascertainable evaluation of the causal relationship [46]. However, the following arguments could be helpful:Extrinsic imputability: A literature search should be performed to identify similar cases.Intrinsic imputability with the following two criteria: (i) Chronological score: ICI-related renal complications have a long latency period. In a series by Cortazar et al., the median (interquartile range) time from immune checkpoint inhibitor initiation to AKI was 14 (6–37) weeks [22] (as opposed to 4 weeks for skin diseases and 6 weeks for colitis). Practitioners should bear in mind that renal complications are possible even after the reintroduction of an ICI [24]. If a rechallenge is performed and AKI occurs again, the score is higher. (ii) Semiological score: Firstly, the patient exhibits known risk factors that have been previously established. A lower baseline eGFR, proton pump inhibitor use, and combination immune checkpoint inhibitor therapy were each independently associated with an increased risk of immune checkpoint inhibitor-associated AKI in the largest series [22]. The following other risk factors should be assessed: pembrolizumab and liver disease [47], as well as age > 65 years. Secondly, the patient experiences or has recently experienced extrarenal irAEs in 40–87% of the cases (hypereosinophilia [48]; immune thrombocytopenic purpura [49]).

On the contrary, the following arguments establish that the diagnosis of ICI-related renal complication requires confirmation: Previous exposure to platinum, pemetrexed [50], iodinated contrast, or a bisphosphonate. When an anti-VEGF has been used before the ICI, the interpretation of proteinuria kinetics is of major importance. Another possible bias is concomitant adrenalitis and adrenal insufficiency [51], leading to pre-renal AKI.

For renal transplant patients, distinctive features are testing for anti-HLA antibodies and BK virus nephropathy. ICIs could lead to very early graft rejection, graft intolerance syndrome, as well as cytokine storm, requiring graft nephrectomy [52] A kidney biopsy is also essential for diagnosis. A recent systematic review of twenty-seven articles with a total of 44 kidney transplant patients treated with ICI, reported a rejection rate of 40.9% [53]. The median time from ICI to a diagnosis of acute rejection was 24 (interquartile range, 10–60) days, which is shorter than the median time reported from ICI to AKI in non-transplant patients. The types of acute allograft rejection reported were cellular rejection (33%), mixed cellular and antibody-mediated rejection (17%), and an unspecified type (50%). The percentage of allograft failure was high (88%), and the mortality rate was 44% [53]. These data are similar to those published in another study that compares the rejection rate in several categories of solid-organ recipients. The highest rejection rate was noted in kidney transplant patients (40.1%), followed by liver (35%) and heart (20%) transplant patients [54]. Recently, a disproportionality analysis of the VigiBase identified drugs associated with rejection. Kidney transplant rejection was associated with nivolumab (IC025 = 1.32), pembrolizumab (IC025 = 1.17), and ipilimumab (IC025 = 0.33), which occurred in the same time frame (21 (interquartile range: 13; 56) days) [55]. In brief, T-cell mediated rejection with low participation of humoral response is the most frequent ICI-related complication in kidney transplant recipients, which is consistent with the suggested pathophysiology of ICI-related breaking of immune tolerance.

## 3. Treatment Strategies

Once it has been established that the renal complication is ICI-related, the treatment strategy should be rapid and efficient because it has now been acknowledged that AKI leads to an increased risk of chronic kidney disease. Whereas a decrease in renal function normally implies morbidity, it means mortality in oncology patients because most subsequent treatments require the highest possible level of renal function. Resolving an irAE can take precedence over the response of the cancer to the ICI. Recent treatment strategies for ICI-related renal complications are summarized in the following paragraph as well as Figure 3.

### 3.1. Stop Exposure to ICI

Although there are no published data on this subject, it is important to know that in life-threatening situations, the use of an antidote should be considered. This could be the case in patients experiencing fulminant myocarditis and AKI, considering that some authors have reported the successful use of plasma exchange (to remove circulating ICIs) and abatacept (to induce a co-stimulation blockade) [56].

In classical situations, supportive care should be initiated (renal replacement therapy: 9% in all ICI-related AKI patients [22], 25% of patients with glomerular diseases [23]). Discontinuation of the ICI as well as suspected associated medications (PPI) is mandatory as soon as an irAE is suspected, to ensure timely implementation of the above-mentioned diagnostic strategy. In most instances, continuing treatment with ICI is not urgent because it is established that the benefit on the tumor lasts even after treatment is discontinued. Assessment by a nephrologist and a kidney biopsy, if performed, are not time-consuming from an oncological point of view. Usually, all aspects can be covered within a week, allowing the ICI to be delayed if necessary.

### 3.2. Stop Immune Response Triggered by ICI

In the prototypical ATIN situation, as well as in numerous irAEs, corticosteroids are the standard of care for ICI-related complications. There is no precise recommended protocol. However, experts agree to advocate 0.8–1 mg/kg/day with a maximum dosage of 60–80 mg per day [57]. Pulse intravenous doses could be administered for 2–3 days, especially if oral absorption is not safe because of associated digestive irAEs. A kidney biopsy should not delay the initiation of corticosteroids. As might be expected, if corticosteroids have been administered for a long period before histological analysis (~>7 days), the hypothesis of a “wash out” of infiltrating cells should be considered. The duration of corticosteroid therapy is not consensual, but a protocol of at least 8–12 weeks is suggested, especially since ICIs can remain bound to the circulating lymphocytes for up to 57 days, with a mean plateau occupancy of 72% [58]. Furthermore, a higher dose of corticosteroids has been associated with a better prognosis [53].

As usual, corticosteroid initiation should take into account the risks of complications (diabetes, infections). Some authors mention the use of PCP prophylaxis with trimethoprim-sulfamethoxazole. However, the risk of introducing a new medication frequently associated with ATIN should be further evaluated. A delayed introduction, when creatinine kinetics are favorable, might be an option to consider.

### 3.3. Tailor Immunosuppression to the Patient

After ICI discontinuation and corticosteroid initiation, it is of the utmost importance to assess the quality of the renal response to first-line treatment. In a series by Cortazar et al., complete, partial, or no kidney recovery occurred in 40%, 45%, and 15% of the patients, respectively. The failure to achieve kidney recovery after immune checkpoint inhibitor-associated AKI was independently associated with higher mortality. Concomitant extrarenal immune-related adverse events were associated with a worse renal prognosis, whereas concomitant tubulointerstitial nephritis-causing medications and treatment with corticosteroids were each associated with improved renal prognosis [22]. For patients with glomerular diseases, renal replacement therapy (RRT) was required in 25% of cases. Most patients had a full (31%) or partial (42%) recovery from an AKI, although 19% remained dialysis-dependent and approximately one-third died. A complete or partial remission of proteinuria was achieved in 45 and 38% of the patients, respectively [23].

#### 3.3.1. Refractory Patients

In an era of evidence-based medicine and omics, we would like to tailor the treatment of ICI-related renal complications to the patient’s individual situation. If a cytokine measurement revealed an associated cytokine increase, specific therapies should be considered (tocilizumab in the case of IL-6 increase). If the ICI dosages indicate high circulating levels, plasma exchanges can be initiated. A kidney biopsy should be performed again to provide additional indications to move to target therapies [59], especially since failure to achieve kidney recovery after an ICI-associated AKI is independently associated with a higher mortality.

#### 3.3.2. Steroid-Dependent Patients or Patients with an Intolerance to Prolonged Steroid Schemes

For these patients, a sparing regimen has to be determined. However, very few data are available for renal irAEs. Data on mycophenolic acid are controversial: deleterious in patients with ATIN leading to pancytopenia and fatal septic complications [60], beneficial in patients with FSGS [61]. Rituximab should have a place in the treatment of ICI-induced vasculitis, as recently reported [62]. Anti-TNF alpha drugs are widely used in digestive irAEs, but there are no indications for renal complications. Future studies are needed to define a clear second-line strategy for patients with complicated ICI-related ATIN.

### 3.4. Follow the Patient

Due to their mechanisms of action, ICI-induced delayed immune responses explain delayed renal complications. The risk of relapse is present, but the incidence has not yet been quantified [63]. In some cases, we can hypothesize that the patient is still exposed to an immune trigger (co-medication). Attention should be paid to the occurrence of the other irAEs, and the patient should be carefully examined every month.

### 3.5. Prevent Relapse If ICI Must Be Re-Started: The “Rechallenge”

If ICIs are the only therapeutic option, the occurrence of a renal complication cannot prevent the rechallenge. As might be expected, these therapeutic choices are subjected to a consultation with nephrologists, oncologists, and the patient. The grade and history of the irAEs are of great importance (for example, in case of ICI-induced myocarditis a rechallenge is strictly forbidden). In a series by Cortazar et al. concerning ICI-related AKI, the rechallenge occurred in 22% of the patients, 23% of whom developed recurrent associated AKI. Forty percent received corticosteroids (prednisone, 10–20 mg daily) in parallel. No data are available on class switching. A key point for the rechallenge is the identification of any drug associated with the first AKI episode. In fact, the prognosis is better if the co-medication has been interrupted.

### 3.6. Prevent the Disease in Future Patients

Predictive factors for renal complications after ICIs are still lacking. It is unknown whether it might be related to the type of malignancy. Before ICI initiation, it is recommended to perform a precise renal examination with data on urinalysis and eGFR, avoid PPI use, and plan a precise follow-up for early detection of any renal complications. In case of solid organ transplant recipients (SOT), minimization of calcineurin inhibitors (CNIs) and the conversion of CNI to mTOR inhibitors (imTORs) along with judicious use of prophylactic steroids could enable the safe use of ICIs in patients with advanced cutaneous squamous cell carcinoma [64]. Clinicians should also be aware of the possibility of renal graft rejection, even in failed allografts [65].

## 4. Conclusions

The renal complications of ICIs still present numerous challenges for onconephrologists. The aim of having the best level of renal function presupposes early detection and recognition of renal complications, adequate biopsies, and rapid treatment. The impact of ICI-related AKI on the patient’s global outcome remains to be defined.

## Figures and Tables

**Figure 1 diagnostics-11-01187-f001:**
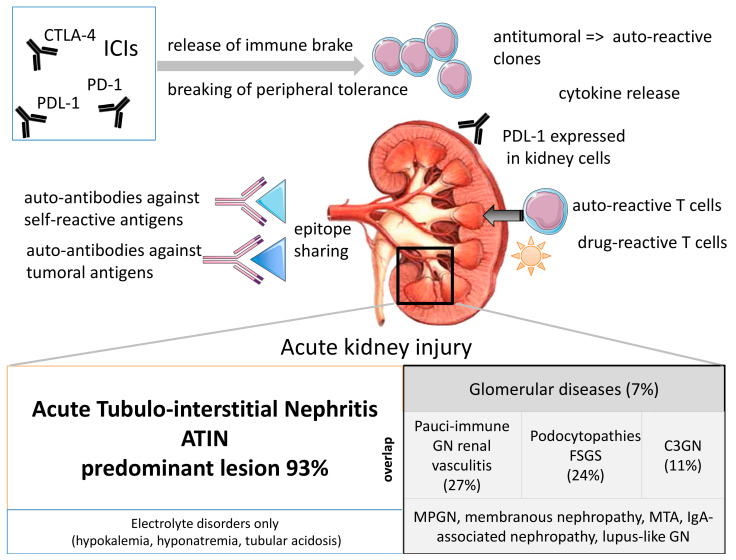
ICI-related renal complications: pathophysiology and landscape. While releasing the immune brake, ICIs lead to a disruption in peripheral tolerance. Through several mechanisms implicating both cellular and humoral immune responses, ICIs may lead to acute kidney injury. The predominant lesion is acute tubulo-interstitial nephritis (ATIN). Some glomerular diseases and electrolyte disorders have also been described.

**Figure 2 diagnostics-11-01187-f002:**
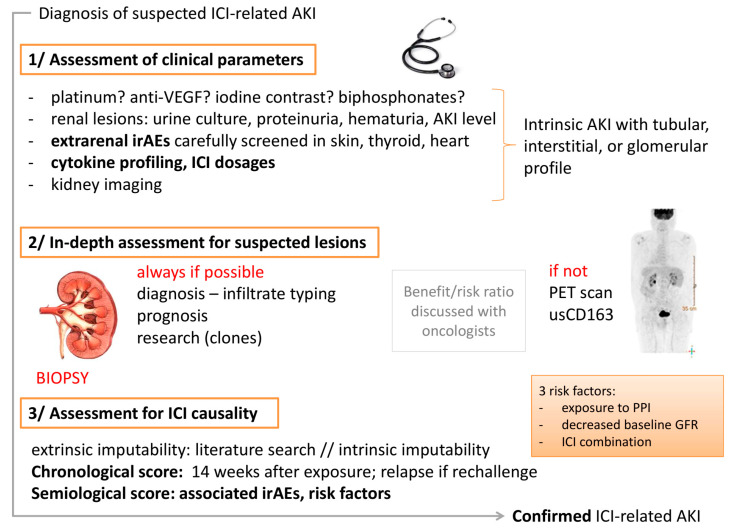
Diagnostic strategy for AKI in a patient exposed to ICIs. When a patient presents with AKI, the clinician should assess whether it is ICI-related. This diagnosis can be difficult, which is why a careful strategy helps to elucidate the causal relationship between the patient’s clinical and biological signs and the exposure to ICIs. PET scan: Positron emission tomography; usCD163: dosage of soluble CD163 level in urine.

**Figure 3 diagnostics-11-01187-f003:**
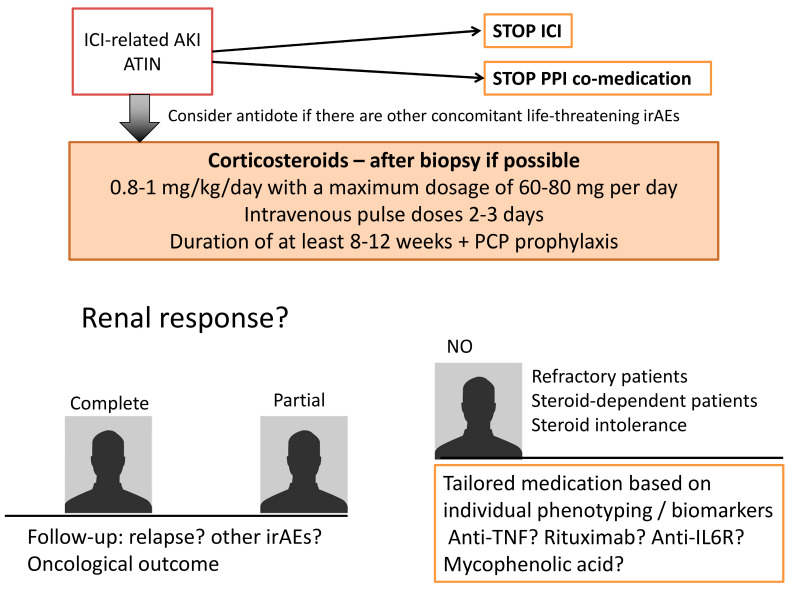
Therapeutic management for ICI-related renal toxicities When a patient is diagnosed with ICI-related ATIN, the therapeutic strategy is based on the discontinuation of ICI and co-medications. Corticosteroids are then the standard of care. Depending on the renal response, tailored medications based on individual phenotyping should be discussed. ATIN: acute tubulointerstitial nephritis; PPI: proton pump inhibitors; irAES immune related adverse events. PCP: pneumocystosis.

**Table 1 diagnostics-11-01187-t001:** ICI list with indications.

ICI Class	Molecule	Date of Approval	Type of Indications
anti-CTLA4	Ipilimumab Tremelimumab Quavonlimab	2011 2015 current folder	melanoma, renal cell carcinoma, CRC mesothelioma, in combination with durvalumab in advanced non-small cell lung cancer, head and neck squamous cell carcinoma, advanced hepatocellular carcinoma
anti-PD1	Pembrolizumab Nivolumab cemiplimab	2014 2014 2018	melanoma, hepatocellular carcinoma, cervical cancer, advanced NSCLC, gastric cancers, Hodgkin lymphoma, primary mediastinal large B-cell lymphoma, urothelial cancer, cutaneous squamous cell carcinoma melanoma, head and neck, hepatocellular carcinoma, renal cell carcinoma, CRC, small lung cancer, advanced NSCLC cutaneous squamous cell carcinoma
anti-PDL1	Atezolizumab Avelumab Durvalumab	2016 2017 2017	advanced small cell lung cancer, advanced NSCLC, triple negative breast cancer, urothelial cancer Merkel cell carcinoma, urothelial cancer urothelial cancer, locally advanced NSCLC, advanced SCLC
anti-LAG3	Eftilagimod alpha Relatlimab	FDA approval March 2020 current folder	metastatic RCC, metastatic breast cancer, melanoma, advanced NSCLC and head and neck squamous cell carcinoma clinical trials recruiting
anti-TIM3	TSR-022 MBG453 Sym023 INCAGN2390 LY3321367 BMS-9862 SHR-170258 RO7121661	current folder	clinical trials recruiting
anti-TIGIT	Tiragolumab MK-7684 Etigilimab BMS-986207 AB-154 ASP-8374	FDA approval January 2021 current folder	PD-L1-high non-small cell lung cancer clinical trials recruiting
anti-VISTA	JNJ-61610588 CA-170	current folder	clinical trials recruiting
anti-B7-H3	Enoblituzumab	FDA approval December 2020	Patients with Pretreated Metastatic HER2-Positive Breast Cancer

CRC: colorectal cancer; NSCLC: Non-small-cell lung carcinoma; SCLC: Small-cell lung carcinoma.

**Table 2 diagnostics-11-01187-t002:** Incidence of renal disorders in individual case safety reports (ICSRs) of suspected adverse drug reactions, according to a recent search in the VigiBase Pharmacovigilance database.

ICI Class	Drug Name	Number of ICSR, *n* =	Number of ICSR with Renal or Urinary Adverse Effects *n* = (%)
anti-CTLA4	Ipilimumab (alone or in combination)	22,641	1021 (4.5%)
	Ipilimumab (combined with nivolumab)	11,536	686 (5.9%)
	tremelimumab	408	22 (5.4%)
anti-PD1	pembrolizumab	29,633	1397 (4.7%)
	nivolumab	51,705	2350 (4.5%)
	cemiplimab	655	50 (7.6%)
anti-PDL1	atezolizumab	8193	431 (5.3%)
	avelumab	1300	82 (6.3%)
	durvalumab	4372	116 (2.7%)
Anti-LAG3 *	relatlimab	65	5 (7.7%)
anti-TIGIT	Tiragolumab	8	0 (0%)
anti-B7-H3	Enoblituzumab	2	0 (0%)

* eftilagim, no ICSR reported in Vigibase as of 11 June 2021.

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
