# Peer review of "Renal Complications Related to Checkpoint Inhibitors: Diagnostic and Therapeutic Strategies"

_diagnostics, 2021, doi:10.3390/diagnostics11071187_

Round 1

Reviewer 1 Report

I thank the authors for this manuscript.
The writing is synthetic making it easy to read. The illustrations are very clear, useful. The structure of the manuscript follows a logic that is easy to follow and consistent with medical exercise. The Vigibase PV analysis is interesting.
The references are well chosen.
It is necessary to modify the paragraph lines 174-176 where ACE and ARB are classified in nephrotoxic drugs in the same way as NSAIDs.

Author Response

Referees' comments:

Referee: 1

Comments to the Author

I thank the authors for this manuscript. The writing is synthetic making it easy to read. The illustrations are very clear, useful. The structure of the manuscript follows a logic that is easy to follow and consistent with medical exercise. The Vigibase PV analysis is interesting. The references are well chosen.

It is necessary to modify the paragraph lines 174-176 where ACE and ARB are classified in nephrotoxic drugs in the same way as NSAIDs.

We thank the referee for his comments and constructive remarks. We agree with the fact that ACE and ARB should not be considered as nephrotoxic drugs. The sentence has been modified in the revised version of the manuscript as follows: “The patient has to be precisely questioned on recent events such as contrast CT scan, nephrotoxic drug exposure (non-steroidal anti-inflammatory drugs, proton pump inhibitors, biphosphonates), dehydration factors, as well as the use of treatments such as angiotensin-converting enzyme inhibitor, angiotensin receptor blockers. A systemic disorder screening should be performed: Sjögren’s syndrome, Raynaud’s syndrome, arthralgia, fever, digestive troubles, chest pain, urinary troubles. “

Reviewer 2 Report

The authors reviewed the clinical features, pathophysiology, diagnostics and therapeutic considerations for immune-checkpoint inhibitor-associated kidney injury. The writing is difficult to follow, mainly due to the language barrier and writing structure, and it doesn’t stand out from other numerous reviews published in this topic. Additionally, it contains numerous flaws, listed (but not limited to) below.

  1. The terminologies are not accurate throughout the manuscript. Please revise. For example “rupture of peripheral tolerance”, “adverse effect” should read “adverse events”.
  2. Introduction, line 29: it’s been 10 years since ICIs have been approved and it’s not “recent” anymore. Please revise.
  3. Scheme 1: please review the indications again, as some of the indications are not updated. For example, pembrolizumab has been approved for cutaneous cell carcinoma as well.
  4. Table 2: I am unsure if reporting the proportion of renal or urinary adverse events in ISCR is meaningful. This may not necessarily reflect the overall incidence the renal or urological adverse events.
  5. Line 84: the pharmacokinetics and the mechanism of distribution lacks references. If these are shared mechanism of distribution among monoclonal antibodies, it may not be worth highlighting here as a characteristics of ICIs.
  6. Reference 8: lacks the journal name (Frontiers Immunol)
  7. Figure 1: the “ICIs” has biohazard marks on their Fc portions. They are not conjugated with cytotoxic agents and this is misleading.
  8. Line 183: urine eosinophils are not recommended as a diagnostic tool for ATIN.
  9. Line 183: “Uro culture” should read urine culture.
  10. Line 187: “Cystatin dosage” should read cystatin C measurement.
  11. Organic AKI should be changed to other wordings, such as intrinsic AKI etc.
  12. Line 221-224: optical microscopy should read "light microscopy". How do you screen for T cell clones in the biopsy? Does this mean kappa/lambda restriction?
  13. Line 277: would expand more on the renal transplant patients and ICI-related complication. There has been a multi-center observational study and meta-analysis on this topic.

Author Response

Referee: 2

Comments to the Author

The authors reviewed the clinical features, pathophysiology, diagnostics and therapeutic considerations for immune-checkpoint inhibitor-associated kidney injury. The writing is difficult to follow, mainly due to the language barrier and writing structure, and it doesn’t stand out from other numerous reviews published in this topic. Additionally, it contains numerous flaws, listed (but not limited to) below.

First, we would like to thank the second referee for his/her review and wise comments. We will try to answer as clearly as possible to your comments and question about this article.

  1. The terminologies are not accurate throughout the manuscript. Please revise. For example “rupture of peripheral tolerance”, “adverse effect” should read “adverse events”.

We thank the referee for his constructive remarks. This has been corrected in the revised version of the manuscript. « Rupture of peripheral tolerance » has been replaced by « Breakdown of peripheral tolerance » and « adverse effect » replaced by « adverse events » when noticed.

  1. Introduction, line 29: it’s been 10 years since ICIs have been approved and it’s not “recent” anymore. Please revise.

We thank the referee for his careful reading. We agree with the referee’s comment. « Recent » has been removed and the sentence modified as follows : « Immune checkpoint inhibitors (ICIs) have been approved in the field of oncology, providing an original antitumor approach compared to chemotherapies. »

  1. Scheme 1: please review the indications again, as some of the indications are not updated. For example, pembrolizumab has been approved for cutaneous cell carcinoma as well.

Table 1 has been updated as suggested by the referee. We also added a sentence on the combination of anti-CTLA4 tremelimumab with anti-PDL1 durvalumab : (i) in advanced non-small cell lung cancer(1); (ii) in head and neck squamous cell carcinoma (2) and (iii) in other solid tumors such as advanced hepatocellular carcinoma (3). Another anti-CTLA4 quavonlimab (MK-1308) has also been reported in combination with pembrolizumab in first-line advanced non-small-cell lung cancer (4) and advanced small-cell lung cancer (5). The corresponding references have been added.

  1. Table 2: I am unsure if reporting the proportion of renal or urinary adverse events in ISCR is meaningful. This may not necessarily reflect the overall incidence the renal or urological adverse events.

We agree with the referee's remark. Indeed, the summary of notifications is different from the frequency of occurrence of adverse drugs reactions evaluated in clinical trials. We propose to clearly specify this element in the legend of the table by the following paragraph :

  • “The distribution of ICSRs is based on pharmacovigilance notifications sent by practitioners or patients. The frequencies in the table are therefore different from the overall incidence of adverse drugs reactions evaluated during clinical trials. The percentages in the table are used to assess the adverse drugs reaction profile of each drug.”

  1. Line 84: the pharmacokinetics and the mechanism of distribution lacks references. If these are shared mechanism of distribution among monoclonal antibodies, it may not be worth highlighting here as a characteristics of ICIs.

We thank the referee for his constructive remarks. Two additional references have been added and the paragraph shortened. We also propose to delete the column concerned by half-lives in the Figure 1. We propose to add the following sentence in the text: The half-life of ICIs are also quite long (vary between 6-27 days) and affected by immune system determinants that increase interindividual variability (6).

  1. Reference 8: lacks the journal name (Frontiers Immunol)

We thank the referee for his careful reading, the reference has been corrected.

  1. Figure 1: the “ICIs” has biohazard marks on their Fc portions. They are not conjugated with cytotoxic agents and this is misleading.

We agree with the referee. Figure 1 has been modified according to his suggestion.

  1. Line 183: urine eosinophils are not recommended as a diagnostic tool for ATIN.

We agree with the referee. This sentence has been removed from the paragraph.

  1. Line 183: “Uro culture” should read urine culture.

We agree with the referee. « Uroculture » has been replaced by « urine culture ».

  1. Line 187: “Cystatin dosage” should read cystatin C measurement.

We agree with the referee. « Cystatin dosage » has been replaced by « cystatin C measurement».

  1. Organic AKI should be changed to other wordings, such as intrinsic AKI etc.

We agree with the referee. « organic AKI» has been replaced by « intrinsic AKI», as well as « obstructive » by « post-renal » AKI.

  1. Line 221-224: optical microscopy should read "light microscopy". How do you screen for T cell clones in the biopsy? Does this mean kappa/lambda restriction?

We agree with the referee : « optical microscopy» has been replaced by « light microscopy». T cell clones can be suspected thanks to membrane surface markers analysis (CD4, CD8) and the expertise of the pathologist on the monomorph or polymorph aspect of the infiltrate. However, the confirmation of T cell clones will come from a T-cell receptor gene rearrangement study. Expensive approaches such as ImmunoSEQ technologies can be considered. Kappa/lambda restriction orient more towards B cell clones. These points have been added to the revised manuscript.

  1. Line 277: would expand more on the renal transplant patients and ICI-related complication. There has been a multi-center observational study and meta-analysis on this topic.

We agree with the referee. The corresponding paragraph has been modified as suggested :

  • « For renal transplant patients, singularities are the search for anti-HLA antibodies and BK virus nephropathy. ICI could lead to very early, brutal and massive graft rejection, intolerance graft syndrome, as well as cytokine storm requiring transplantectomy (Belliere et al., AJT, in press). Kidney biopsy is also essential for diagnosis. As recently reported in a systematic review, twenty-seven articles with a total of 44 kidney transplant patients treated with ICI were identified, with a rejection rate of 40.9% (7). Median time from ICI to acute rejection diagnosis was 24 (interquartile range, 10–60) days, which is shorter than the median reported time from ICI to AKI in non-transplant patients. Reported types of acute allograft rejection were cellular rejection (33%), mixed cellular and antibody-mediated rejection (17%), and unspecified type (50%). Percentage of allograft failure was high (88%) and mortality rate was 44%(7). These data are similar to that published in another study comparing rejection rate in several categories of solid-organ recipients: the highest rejection rate was seen in patients with kidney transplants (40.1%), then liver (35%) and heart (20%) transplants(8). Recently, a disproportionality analysis of the VigiBase identified drugs associated with rejection events: kidney transplant rejection was associated with nivolumab (IC025 = 1.32), pembrolizumab (IC025 = 1.17) and ipilimumab (IC025 = 0.33), occurring in the same time frame (21 [interquartile range: 13; 56] days) (9). To summarize, T-cell mediated rejection with low participation of humoral response is the most frequent ICI-related complication in kidney transplant recipient, consistent with the suggested pathophysiology of ICI-related breaking of immune tolerance. »

References

  1. Rizvi NA, Cho BC, Reinmuth N, Lee KH, Luft A, Ahn MJ, Van Den Heuvel MM, Cobo M, Vicente D, Smolin A, Moiseyenko V, Antonia SJ, Le Moulec S, Robinet G, Natale R, Schneider J, Shepherd FA, Geater SL, Garon EB, Kim ES, Goldberg SB, Nakagawa K, Raja R, Higgs BW, Boothman AM, Zhao L, Scheuring U, Stockman PK, Chand VK, Peters S: Durvalumab with or Without Tremelimumab vs Standard Chemotherapy in First-line Treatment of Metastatic Non-Small Cell Lung Cancer: The MYSTIC Phase 3 Randomized Clinical Trial. JAMA Oncol 6: 661–674, 2020
  2. Wang B-C, Li P-C, Fan J-Q, Lin G-H, Liu Q: Durvalumab and tremelimumab combination therapy versus durvalumab or tremelimumab monotherapy for patients with solid tumors. Medicine (Baltimore) 99: e21273, 2020
  3. Kelley R, Kudo M, Harris W, Ikeda M, Okusaka T, Kang Y, Qin S, Tai D, Lim H, Yau T, Yong W, Cheng A, Gasbarrini A, de Braud F, Bruix J, Borad M, Standifer N, He P, Negro A, Vlahovic G, Sangro B, Abou-Alfa G: O-6 The novel regimen of tremelimumab in combination with durvalumab provides a favorable safety profile and clinical activity for patients with advanced hepatocellular carcinoma. Ann Oncol 31: 233–234, 2020
  4. Perets R, Bar J, Rasco DW, Ahn MJ, Yoh K, Kim DW, Nagrial A, Satouchi M, Lee DH, Spigel DR, Kotasek D, Gutierrez M, Niu J, Siddiqi S, Li X, Cyrus J, Chackerian A, Chain A, Altura RA, Cho BC: Safety and efficacy of quavonlimab, a novel anti-CTLA-4 antibody (MK-1308), in combination with pembrolizumab in first-line advanced non-small-cell lung cancer. Ann Oncol 32: 2021
  5. Cho BC, Yoh K, Bar J, Nagrial A, Spigel DR, Gutierrez M, Kim D-W, Kotasek D, Rasco D, Niu J, Satouchi M, Ahn M-J, Lee DH, Maurice-Dror C, Siddiqi S, Li X, Cyrus J, Altura RA, Perets R: Results From a Phase I Study of MK-1308 (ANTI–CTLA-4) Plus Pembrolizumab in Previously Treated Advanced Small Cell Lung Cancer. Ann Oncol 30: xi36–xi37, 2019
  6. Keizer RJ, Huitema ADR, Schellens JHM, Beijnen JH: Clinical pharmacokinetics of therapeutic monoclonal antibodies. Clin. Pharmacokinet. 49: 493–507, 2010
  7. Manohar S, Thongprayoon C, Cheungpasitporn W, Markovic SN, Herrmann SM: Systematic Review of the Safety of Immune Checkpoint Inhibitors Among Kidney Transplant Patients. Kidney Int Reports 5: 149–158, 2020
  8. Fisher J, Zeitouni N, Fan W, Samie FH: Immune checkpoint inhibitor therapy in solid organ transplant recipients: A patient-centered systematic review. J. Am. Acad. Dermatol. 2019
  9. Nguyen LS, Ortuno S, Lebrun-Vignes B, Johnson DB, Moslehi JJ, Hertig A, Salem JE: Transplant rejections associated with immune checkpoint inhibitors: A pharmacovigilance study and systematic literature review. Eur J Cancer 148: 36–47, 2021

Reviewer 3 Report

  • To make the list of immune check point inhibitors more comprehensive, I would recommend to review this article and add the novel targets in the table 1. This will make the list uptodate. 

https://molecular-cancer.biomedcentral.com/articles/10.1186/s12943-019-1091-2 

  • Page 5: ICIs present extrarenal clearance: please reframe this sentence. Also add a reference for this.
  • Page 6: Other glomerular lesions have been observed (26): IgA-associated glomerulonephritis (GN) (27), Goodpasture syndrome (28), membranoproliferative GN (29), lupus-like nephropathy (15), and thrombotic micro-angiopathy (30). Replace the term membranoproliferative GN with membranoproliferative GN pattern.
  • Page 6: Of course, an overlap between ATIN and glomerular diseases can be noticed (Figure 1). Interestingly, in some patients, renal lesions are only revealed by electrolyte disorders, including hyponatremia secondary to hypophysitis, hypokalemia (31), and distal renal tubular acidosis: Please do not start a new sentence with words like “Of course”. Reframe the sentence.

  • Page 7: The predominant lesion is acute tubule-interstitial nephritis (ATIN). Some glomerular diseases and electrolyte disorders have been described as well. : Please change it to tubulo-interstitial nephritis.

  • Page 7: As mentioned before, isolated electrolyte disorders can be observed, as well as isolated urinalysis abnormalities (at the early phase of podocyte injury, for example). Please reframe the sentence.

  • Page 8: In some oncology guidelines, when serum creatinine increases by a factor of 1.5-2, intervention is not indicated. Not sure where this has come from. Please provide a reference. This statement seems incorrect. 1.5 to 2 times rise in Sr creatinine is a very significant change.

  • Page 8: As usual, the diagnosis strategy includes a precise screening for patient medical history (nephrological and oncological aspects), current and previous medications, as well as cardiovascular risk factors and habits.: Please do not begin a sentence with As usual. Reframe the sentence.

  • Page 9: Biological exams will encompass urinalysis to assess for leukocyturia, hematuria, urine culture, as well as sodium/potassium ratio, magnesium, and sodium excretion fraction calculation, and of course proteinuria, micro-albuminuria, and creatininuria measurement.: remove the word “of course.”

  • Page 9: Beyond sodium, potassium, bicarbonates, urea, and creatinine dosage, magnesium and uric acid fraction can be useful in the characterization of tubular lesions. : Explain in what way it is useful.

  • Page 9: Whereas international guidelines do not recommend discussing kidney biopsy as a first-line treatment (35) (36), nephrologists are fighting against this practice.: grammatically incorrect framing. Also kidney biopsy is an investigative tool and not a treatment.

  • Page 9: This underlines the importance of proving ATIN to avoid senseless exposure to steroids.: do not use words like senseless. This is not scientifically correct. And actually, treatment for acute interstitial nephritis is steroids. Do the authors want to say ATN instead of ATIN ?

  • Page 9: Light microscopy as well as immunofluorescence should be systematic. : What is the meaning of this sentence. Need to reframe.

  • Page 9: T cell clones can be suspected thanks to membrane surface markers analysis (CD4, CD8) and the expertise of the pathologist on the monomorphic or polymorph aspect of the infiltrate. : reframe this sentence. Don’t not use words like thanks.

  • Page 10: Of course, if kidney biopsy is not possible, some non-invasive markers have been studied in preliminary works.: reframe sentence. Do not start new sentence with words like of course.

  • Page 11: In various AKI conditions, suCD163 has also been identified as a relevant marker: prospective studies are needed to validate its role in ICI-related AKI, especially when invasive procedure is not possible: Reference? Also, what does author mean by various AKI conditions.

  • Page 11: ICI could lead to very early, brutal and massive graft rejection, intolerance graft syndrome, as well as cytokine storm requiring transplantectomy : avoid using adjectives like brutal or massive. Reframe the sentence. Translantectomy is not a scientific terminology. Can use graft nephrectomy instead.

Author Response

Reviewer 3

  • To make the list of immune check point inhibitors more comprehensive, I would recommend to review this article and add the novel targets in the table 1. This will make the list uptodate. 

https://molecular-cancer.biomedcentral.com/articles/10.1186/s12943-019-1091-2 

We thank the reviewer for his suggestion. We added the corresponding reference and updated Table 1.

  • Page 5: ICIs present extrarenal clearance: please reframe this sentence. Also add a reference for this.

This has been done.

  • Page 6: Other glomerular lesions have been observed (26): IgA-associated glomerulonephritis (GN) (27), Goodpasture syndrome (28), membranoproliferative GN (29), lupus-like nephropathy (15), and thrombotic micro-angiopathy (30). Replace the term membranoproliferative GN with membranoproliferative GN pattern.

This has been done.

  • Page 6: Of course, an overlap between ATIN and glomerular diseases can be noticed (Figure 1). Interestingly, in some patients, renal lesions are only revealed by electrolyte disorders, including hyponatremia secondary to hypophysitis, hypokalemia (31), and distal renal tubular acidosis: Please do not start a new sentence with words like “Of course”. Reframe the sentence.

This has been done.

  • Page 7: The predominant lesion is acute tubule-interstitial nephritis (ATIN). Some glomerular diseases and electrolyte disorders have been described as well.: Please change it to tubulo-interstitial nephritis.

This has been done.

  • Page 7: As mentioned before, isolated electrolyte disorders can be observed, as well as isolated urinalysis abnormalities (at the early phase of podocyte injury, for example). Please reframe the sentence.

This has been done.

  • Page 8: In some oncology guidelines, when serum creatinine increases by a factor of 1.5-2, intervention is not indicated. Not sure where this has come from. Please provide a reference. This statement seems incorrect. 1.5 to 2 times rise in Sr creatinine is a very significant change.

We agree with the reviewer and apologize for the unclear sentence. We meant that a slight increase in creatinine level should raise the attention of the clinicians and lead them to refer the patient to a nephrologist, even if the rise in Sr creatinine is less than 1.5. The guidelines we mentioned came from the following publication: “Management of Immune-Related Adverse Events in Patients Treated with Immune Checkpoint Inhibitor Therapy: American Society of Clinical Oncology Clinical Practice Guideline, 2018”. The revised version has been modified and the sentence corrected.

  • Page 8: As usual, the diagnosis strategy includes a precise screening for patient medical history (nephrological and oncological aspects), current and previous medications, as well as cardiovascular risk factors and habits.: Please do not begin a sentence with As usual. Reframe the sentence.

 This has been done.

  • Page 9: Biological exams will encompass urinalysis to assess for leukocyturia, hematuria, urine culture, as well as sodium/potassium ratio, magnesium, and sodium excretion fraction calculation, and of course proteinuria, micro-albuminuria, and creatininuria measurement.:remove the word “of course.”

  This has been done.

  • Page 9: Beyond sodium, potassium, bicarbonates, urea, and creatinine dosage, magnesium and uric acid fraction can be useful in the characterization of tubular lesions.: Explain in what way it is useful.

We agree with the reviewer and modified the manuscript as follow: “Given that some cases of ICI-related kidney toxicity may be restricted to isolated electrolyte disorders, clinicians should be aware of small variations in routine lab tests which suggest tubular dysfunction”.

  • Page 9: Whereas international guidelines do not recommend discussing kidney biopsy as a first-line treatment (35) (36), nephrologists are fighting against this practice.: grammatically incorrect framing. Also kidney biopsy is an investigative tool and not a treatment.

We agree with the reviewer and apologize for this mistake. The sentence has been corrected.

  • Page 9:This underlines the importance of proving ATIN to avoid senseless exposure to steroids.: do not use words like senseless. This is not scientifically correct. And actually, treatment for acute interstitial nephritis is steroids. Do the authors want to say ATN instead of ATIN?

We agree with the reviewer and apologize for this mistake. The sentence has been corrected as follows: “This underlines the importance of proving acute tubular necrosis without an inflammatory component to avoid exposure to steroids.”

  • Page 9: Light microscopy as well as immunofluorescence should be systematic.: What is the meaning of this sentence. Need to reframe.

 We agree with the reviewer and apologize for this unclear sentence. We meant that kidney biopsy should be as representative as possible (including fixed and frozen sections, with a sufficient number of glomeruli) to allow routine staining and detection of antibodies.

Page 9: T cell clones can be suspected thanks to membrane surface markers analysis (CD4, CD8) and the expertise of the pathologist on the monomorphic or polymorph aspect of the infiltrate.: reframe this sentence. Don’t not use words like thanks.

 This has been changed. The word “thanks” has been replaced by “considering”. The following sentence has been added: “The pathologist should be able to specify whether infiltrate is monomorphic or polymorphic.”

  • Page 10: Of course, if kidney biopsy is not possible, some non-invasive markers have been studied in preliminary works.: reframe sentence. Do not start new sentence with words like of course.

 This has been changed.

  • Page 11: In various AKI conditions, suCD163 has also been identified as a relevant marker: prospective studies are needed to validate its role in ICI-related AKI, especially when invasive procedure is not possible: Reference? Also, what does author mean by various AKI conditions.

We added 2 references that assess suCD163 use in vasculitis and lupus nephropathies. The sentence has been changed according to the reviewer’s suggestion.

  • Page 11: ICI could lead to very early, brutal and massive graft rejection, intolerance graft syndrome, as well as cytokine storm requiring transplantectomy: avoid using adjectives like brutal or massive. Reframe the sentence. Translantectomy is not a scientific terminology. Can use graft nephrectomy instead.

We agree with the reviewer: the adjectives have been removed and the terminology corrected.

Round 2

Reviewer 2 Report

The writing is still difficult to follow, mainly due to the language barrier and writing structure, and it doesn’t stand out from other numerous reviews published in this topic. Additionally, it contains numerous flaws, which did not significantly improve after the edits that were made. 

Author Response

As suggested by the reviewer, the manuscript has been entirely corrected by an American native speaker. The certificate will be attached to the revision.

Reviewer 3 Report

Accept